# Rethinking remdesivir for COVID-19: A Bayesian reanalysis of trial findings

**Joyce M. Hoek**⊙*, **Sarahanne M. Field**⊙, **Ymkje Anna de Vries, Maximilian Linde, Merle-Marie Pittelkow, Jasmine Muradchanian, Don van Ravenzwaaij**

Behavioural and Social Sciences, University of Groningen, Groningen, The Netherlands

⊙ These authors contributed equally to this work.
* j.m.hoek@rug.nl

## Abstract

### Background

Following testing in clinical trials, the use of remdesivir for treatment of COVID-19 has been authorized for use in parts of the world, including the USA and Europe. Early authorizations were largely based on results from two clinical trials. A third study published by Wang et al. was underpowered and deemed inconclusive. Although regulators have shown an interest in interpreting the Wang et al. study, under a frequentist framework it is difficult to determine if the non-significant finding was caused by a lack of power or by the absence of an effect. Bayesian hypothesis testing does allow for quantification of evidence in favor of the absence of an effect.

### Findings

Results of our Bayesian reanalysis of the three trials show ambiguous evidence for the primary outcome of clinical improvement and moderate evidence against the secondary outcome of decreased mortality rate. Additional analyses of three studies published after initial marketing approval support these findings.

### Conclusions

We recommend that regulatory bodies take all available evidence into account for endorsement decisions. A Bayesian approach can be beneficial, in particular in case of statistically non-significant results. This is especially pressing when limited clinical efficacy data is available.

## Introduction

In March 2020, the World Health Organization (WHO) declared the COVID-19 pandemic. A year later, the SARS-CoV-2 virus has infected over 127 million people globally and over 2.7 million people have died [1]. The pandemic has set an unprecedented challenge for

**Data Availability Statement:** All relevant data are within the manuscript and its Supporting information files. In addition, all data can be found

on the OSF page of the project at DOI 10.17605/OSF.IO/KDQT3.

**Funding:** This research was supported by a Dutch scientific organization VIDI fellowship grant (016. Vidi.188.001, http://www.nwo.nl) to DvR. The funders had no role in study design, data collection and analysis, decision to publish, or preparation of the manuscript.

**Competing interests:** The authors have declared that no competing interests exist.

pharmaceutical companies, and laboratories around the world are racing to develop treatments, both for management and prevention. Remdesivir, a broad-spectrum antiviral, was one of the first major contenders in the race. Promising clinical trial results released on April 29, 2020 led to remdesivir receiving emergency use authorization for treatment of COVID-19 in the United States on May 1 [2]. It was officially licensed by Japan on May 7 [3], and Europe on July 3 [4] among other countries.

At the time of these initial approvals, results of only two randomized placebo-controlled trials were available: a prematurely terminated and underpowered study by Wang et al., which found no statistically significant effect of remdesivir on time to clinical improvement or mortality rate [5]; and NIAID's ACTT-1 trial, which found a statistically significant shorter time to clinical improvement in the remdesivir group, while the effect for mortality fell just short of statistical significance [6]. In addition, results from Gilead's randomized open-label GS-US-540-5773 trial were available, which showed no significant differences between 5- and 10-day remdesivir treatment [7]. Regulators based their initial approvals of remdesivir mainly on the last two studies. Unable to interpret the results from Wang et al., they failed to include these study outcomes in their efficacy assessment of remdesivir [2, 8].

The trial by Wang et al. was deemed inconclusive because it is difficult to determine, in a frequentist framework, whether the non-significant outcome of this trial is caused by a lack of power or a lack of effect of remdesivir. Despite difficulties in interpreting the results, the European Medicines Agency (EMA) did express interest in including the Wang et al. study in their assessment of remdesivir [9] and described the study in detail in its public assessment report. However, EMA refrained from drawing conclusions about the efficacy of remdesivir based upon this study [8].

Since only two randomized placebo-controlled trials were available at the time of initial approval, it is important that regulators can interpret the results from both of these trials. Especially for interpretation of the Wang et al study, it is crucial to be able to quantify evidence in favor of *either* the alternative (remdesivir works) *or* the null hypothesis (remdesivir does not work more effectively than a placebo). In contrast to NHST, Bayesian statistics can provide such insight, as it allows the user to quantify evidence in favor of efficacy relative to evidence in favor of non-efficacy by means of a Bayes factor [10]. The Bayes factor thus provides a continuous measure of evidence. In the current study, we will showcase the strengths of Bayesian reanalysis for drug trials.

## Methods

In this paper, we provide a Bayesian reanalysis of the three remdesivir studies initially used for approval by the medicine regulators (ACTT-1 preliminary report [6], GS-US-540-5773 [7], and the Wang et al. trial [5]). In addition, we evaluate one updated study (ACTT-1 final report [11]) and two studies that were published after remdesivir was officially approved (GS-US-540-5774 [12] and WHO Solidarity [13]) to provide a more detailed and complete picture of the currently available data on remdesivir. Full details of our analytic approach are included in the S1 and S2 Files, which can also be found at https://osf.io/kdqt3.

Since the available studies use different outcomes to measure the efficacy of remdesivir, a Bayesian meta-analysis is not possible. We therefore analyzed all studies separately. Specifically, we reanalyzed two important outcomes: clinical improvement and mortality rate. Clinical improvement was measured differently in all trials. Wang et al. measured time to clinical improvement within 28 days after randomization as a "two-point reduction in patients' admission status on a six-point ordinal scale, or live discharge, whichever came first" [5]. The ACTT-1 trial measured time to recovery as the first day of "either discharge from the hospital

or hospitalization for infection control purposes only" during 28 days after enrollment [6, 11]. GS-US-540-5773 measured "clinical status on day 14, assessed on a 7-point ordinal scale" [7]. GS-US-540-5774 measured "the distribution of clinical status assessed on the 7-point ordinal scale on study day 11" [12]. The Solidarity trial did not measure clinical improvement [13]. Mortality rate was assessed on day 14 [6, 7, 11] or 28 [5, 11, 13] of the trials.

We used the available summary statistics as reported in the articles to calculate Bayes factors using Bayesian two-sided $t$-tests and Bayesian chi-square tests. Wang et al. reported results for early treatment (<10 days of symptom onset) and the Solidarity trial for treatment in less severely ill patients (no mechanical ventilation), suggesting efficacy might be greater if remdesivir is administered early in the course of illness. When appropriate summary statistics were available, we calculated Bayes factors for these groups as well. Note that the ACTT-1 trial reports results for these early treatment groups, but without adequate summary statistics, we were not able to calculate Bayes factors.

### Bayesian two-sided *t*-test

We performed Bayesian two-sided $t$-tests, calculating a Jeffreys-Zellner-Siow default Bayes factor [14, 15]. Under the alternative hypothesis, the prior distribution for the standardized effect size parameter $\delta$ is a Cauchy distribution centered on 0 with a scale parameter of $\frac{1}{\sqrt{2}}$, meaning that 50% of the probability mass lies between -0.707 and 0.707. In the absence of a-priori indications of efficacy of remdesivir in the treatment of COVID-19, there are clear advantages to using a default prior, such as the Cauchy distribution. The Cauchy prior is often used because it satisfies some desirable mathematical properties, and is relatively easy to specify and interpret. For a detailed discussion about the Jeffreys-Zellner-Siow Bayes factor and the advantage of using a zero-centered Cauchy prior, please refer to van [16, p5]. The analyses were conducted using the R packages BayesFactor [17] and baymedr [18].

### Bayesian chi-square test

In addition, we performed Bayesian chi-square tests, testing the null hypothesis of independent rows/columns against the alternative hypothesis of dependent rows/columns [19]. Under the null hypothesis, the joint mortality rate parameter is uniformly distributed (i.e., Beta (1,1), meaning that all values are equally likely to occur). Under the alternative hypothesis, independent mortality rate parameters for each group are both uniformly distributed (both Beta (1,1)). The analyses were conducted using the R package BayesFactor [17].

As we do not have access to the full data of all trials, we used the summary statistics reported by the authors of the published articles to approximate Bayes factors. Throughout this article, we report Bayes factors for which values greater than 1 indicate evidence for the null hypothesis (no efficacy), while values below 1 indicate evidence for the alternative hypothesis (efficacy) and denote them with $BF_{01}$. A $BF_{01}$ greater than 3 (or less than 1/3) is generally considered moderate evidence, while a $BF_{01}$ greater than 10 (or less than 1/10) is considered strong evidence [20, 21].

## Results

Fig 1 and the S1 Table show the results of our reanalysis.

### Reanalysis of studies used of initial marketing approval of remdesivir

Our reanalysis of the preliminary report of the ACTT-1 trial [6] shows that remdesivir outperforms placebo for time to clinical recovery ($BF_{01}$ = 0.13, i.e., the data is 1/0.13 = 7.7 times more

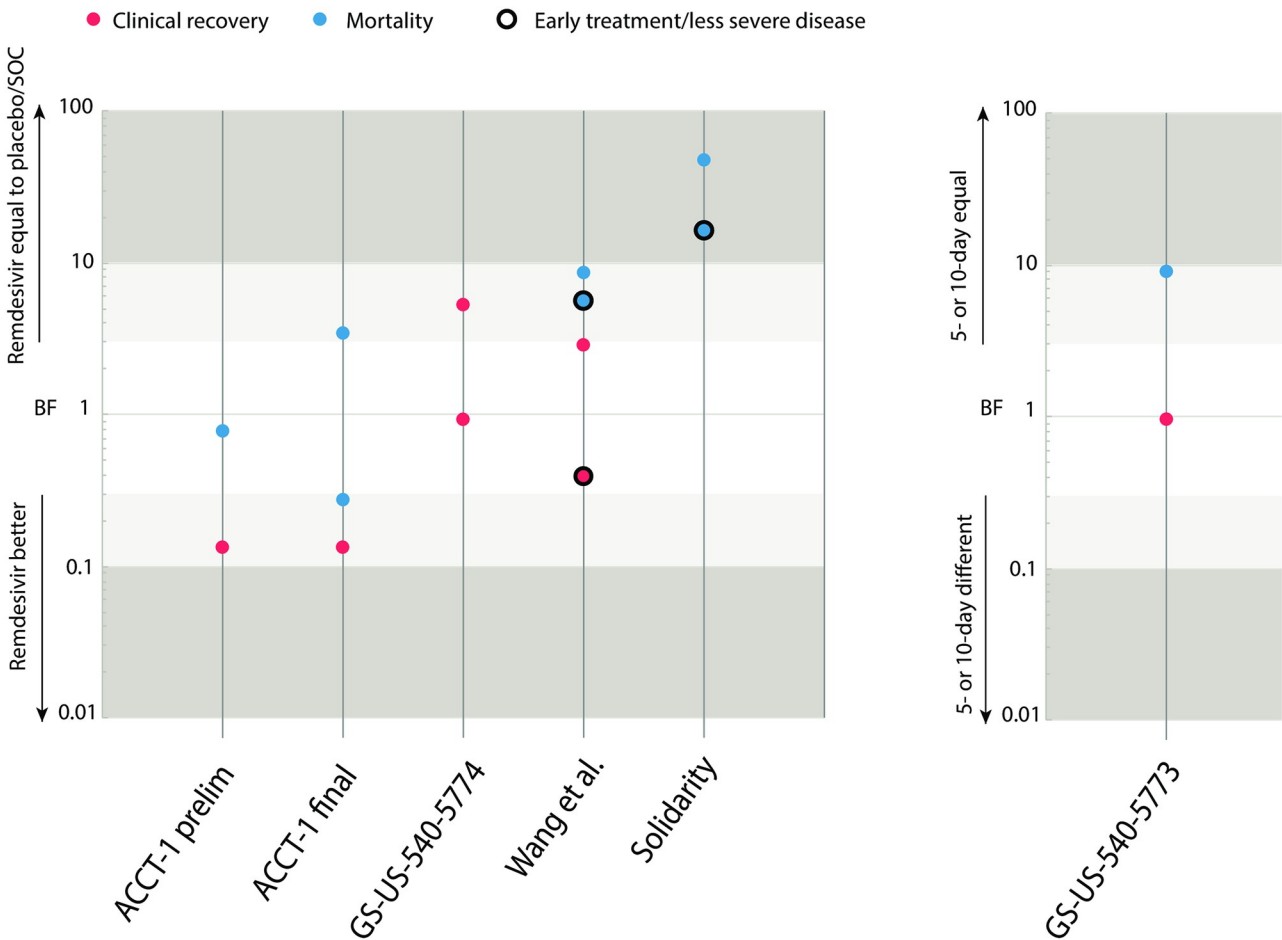

**Fig 1. Bayesian reanalysis of clinical trials testing the efficacy of remdesivir against COVID-19.** A $BF_{01}$ between 1/3 and 3 (white area) indicates ambiguous evidence, above 3 or below 1/3 (light grey area) moderate evidence, and above 10 or below 1/10 (dark grey area) strong evidence. A black circle either indicates treatment within 10 days of symptom onset (Wang et al.) or treatment on patients without mechanical ventilation (Solidarity). The GS-US-540-5773 trial did not compare remdesivir to placebo, but instead showed that there is no difference between a 5- or 10-day regimen of remdesivir.

likely under the alternative hypothesis, indicating moderate evidence pro-remdesivir). However, the evidence that remdesivir outperforms placebo for mortality rate is ambiguous ($BF_{01}$ = 0.75, meaning the data is about equally likely under the alternative or null hypothesis).

In comparison, our reanalysis of Wang et al. [5] provides ambiguous evidence *against* remdesivir-treated patients improving more quickly than patients in the placebo group ($BF_{01}$ = 2.8, i.e., the data is almost three times more likely under the null hypothesis of no effect than under the alternative hypothesis). The reanalysis of the mortality rate data yielded moderate evidence in favor of no effect ($BF_{01}$ = 8.3). Within the subgroup who received remdesivir within 10 days of symptom onset, we report a $BF_{01}$ of 0.38 (data is 1/0.38 = 2.6 more likely under the alternative hypothesis) and 5.4 for the time to clinical improvement and mortality, respectively. Taken together, the evidence from this study points towards non-efficacy of remdesivir.

The GS-US-540-5773 trial compared a 5- to a 10-day treatment of remdesivir. A placebo group was not included in this trial [7]. Our reanalysis of the data largely supports Goldman and colleagues' null hypothesis of no difference between a 5- or 10-day course of remdesivir

($BF_{01} = 0.97$ for the primary outcome, showing equipoise; $BF_{01} = 9.1$ for mortality rate, showing moderate evidence for the null hypothesis). However, note that any investigation into the optimal dosage of remdesivir is predicated on the drug being effective to begin with.

### Reanalysis of studies published after initial marketing approval of remdesivir

After initial marketing approval, the final results of the ACTT-1 trial were published in an updated report [11]. Our reanalysis of this final report shows similar evidence for the efficacy of remdesivir compared to the result from the preliminary report ($BF_{01} = 0.13$, moderate evidence pro-remdesivir). Furthermore, we show moderate evidence that remdesivir outperforms placebo for mortality by day 15 ($BF_{01} = 0.27$), and moderate evidence *against* the efficacy of remdesivir for mortality by day 29 ($BF_{01} = 3.3$).

The GS-US-540-5774 trial compared a 5- or 10-day regimen of remdesivir to standard of care [12]. Results from this trial show ambiguous evidence for a difference in clinical status between the SOC-group and patients on a 5-day course of remdesivir ($BF_{01} = 0.91$) and moderate evidence in favor of no effect for patients on a 10-day course of remdesivir ($BF_{01} = 5.1$).

The randomized open-label Solidarity trial [13] compared a 10-day treatment of remdesivir to standard of care. It showed strong evidence *against* remdesivir improving in-hospital mortality in both the full patient group ($BF_{01} = 45.4$) and in patients receiving no mechanical ventilation ($BF_{01} = 15.8$).

### Discussion

The results of our Bayesian reanalyses of the available studies at time of marketing approval of remdesivir show that these three studies (ACTT-1 preliminary report, the GS-US-540-5773 trial, and the Wang et al. trial) are comparable in terms of evidential strength. In terms of mortality rate, the data of these studies point to moderate pro-null evidence (i.e., remdesivir is no more effective than the placebo), or ambiguous evidence, meaning that an effect of remdesivir on mortality rate is unclear. The data of the Wang et al. trial provide ambiguous evidence for an effect of remdesivir on time to clinical improvement, while the ACTT-1 trial showed moderate pro-efficacy evidence. Importantly, the Wang et al study provides moderate evidence against remdesivir's efficacy when it comes to mortality, which contrasts the ACTT-1 trial. Since the Wang et al. trial is comparable in evidential strength to the ACTT-1 trial, we argue that it would be unwise to ignore this study for authorization decisions.

In addition, we evaluated the results of two new publications and one updated publication that became available after initial marketing authorization. These studies show evidence against the effectiveness of remdesivir when it comes to mortality and provide moderate or ambiguous evidence against the effectiveness of remdesivir on improvement of clinical recovery. Only the final results of the ACTT-1 trial show moderate evidence in support of remdesivir's effectiveness in clinical improvement and mortality at day 15. Notably, similar to mortality results of the other trials, the ACTT-1 day-29 mortality data provides moderate evidence *against* the efficacy of remdesivir.

### Limitations

Our findings need to be interpreted in light of three limitations. Firstly, we did not have access to the raw trial data for any of the studies. Because the provided summary statistics did not always provide all the desired information (e.g., standard deviations), our analyses should be seen as reasonable approximations based on available alternative information (e.g., interquartile ranges).

Secondly, the original analyses were sometimes based on survival analyses. Because Bayesian tests for survival analyses are not well-developed (but see [22] for an overview of Bayesian survival analysis techniques focusing on parameter estimation), we approximated these analyses using Bayesian *t*-tests and confirmed that the *t*-test statistics were of similar magnitude to those obtained under a frequentist survival analysis framework. It must be noted that survival data is often positively skewed, which violates the assumptions of a *t*-test. Although *t*-tests are generally robust to violations of normality, this may have affected the results of our reanalysis.

Finally, all Bayesian analyses were conducted using default priors. We believe that to be reasonable in the absence of strong prior knowledge about the efficacy and safety of remdesivir. Although we always used the same distribution family for the prior, we did conduct sensitivity analyses, varying the scale parameter of the Cauchy prior, to make sure our analysis was robust (see supplemental information). Nevertheless, different distributional choices of priors may be defensible and could have led to different outcomes. Given these limitations, our findings should be interpreted as a first step toward a thorough Bayesian reanalysis of these trials, and we encourage further Bayesian reanalyses in other contexts and for other drug testing based on the individual patient data to ensure that the statistical model is appropriate.

## Conclusion

By reanalyzing the remdesivir studies used to obtain marketing approval, we demonstrate that a complete picture of the evidence for remdesivir as a potential COVID-19 treatment requires that all available clinical data should be taken into account for decision-making processes by regulatory authorities. Furthermore, our Bayesian reanalysis of all six available remdesivir publications provides a more detailed and complete picture of the current state of evidence than is currently available, suggesting that remdesivir has little to no effect against COVID-19. The current paper also demonstrates the value of Bayesian analysis in the interpretation of clinical trial data, especially when applied in cases where NHST cannot be used to assess ambiguous or pro-null evidence.

## Supporting information

**S1 File. Detailed analysis.**
(PDF)

**S2 File. Code of analysis.**
(RMD)

**S1 Table. Overview of Bayes factors of the reanalyzed studies.**
(PDF)

## Author Contributions

**Conceptualization:** Joyce M. Hoek, Ymkje Anna de Vries, Don van Ravenzwaaij.

**Formal analysis:** Don van Ravenzwaaij.

**Funding acquisition:** Don van Ravenzwaaij.

**Investigation:** Joyce M. Hoek, Sarahanne M. Field.

**Methodology:** Don van Ravenzwaaij.

**Project administration:** Joyce M. Hoek, Sarahanne M. Field.

**Software:** Don van Ravenzwaaij.

**Validation:** Ymkje Anna de Vries, Maximilian Linde, Merle-Marie Pittelkow, Jasmine Muradchanian.

**Writing – original draft:** Joyce M. Hoek, Sarahanne M. Field.

**Writing – review & editing:** Joyce M. Hoek, Sarahanne M. Field, Ymkje Anna de Vries, Maximilian Linde, Merle-Marie Pittelkow, Jasmine Muradchanian, Don van Ravenzwaaij.

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
