## [Decision Letter · Decision Letter 0]

2 Feb 2021

PONE-D-20-30134

Rethinking remdesivir for COVID-19: a Bayesian reanalysis of trial findings

PLOS ONE

Dear Dr. Hoek,

Thank you for submitting your manuscript to PLOS ONE. After careful consideration, we have decided that your manuscript does not meet our criteria for publication and must therefore be rejected.

This is because both reviewers expressed many concerns and one of them recommended rejection. I have tried to invite the third reviewer but did not receive any feedback by now. 

I am sorry that we cannot be more positive on this occasion, but hope that you appreciate the reasons for this decision.

Yours sincerely,

Gang Han, PhD

Academic Editor

PLOS ONE

Reviewers' comments:

Reviewer's Responses to Questions

**Comments to the Author**

1. Is the manuscript technically sound, and do the data support the conclusions?

Reviewer #1: Partly

Reviewer #2: Partly

2. Has the statistical analysis been performed appropriately and rigorously? 

Reviewer #1: Yes

Reviewer #2: No

3. Have the authors made all data underlying the findings in their manuscript fully available?

Reviewer #1: No

Reviewer #2: Yes

4. Is the manuscript presented in an intelligible fashion and written in standard English?

Reviewer #1: Yes

Reviewer #2: Yes

5. Review Comments to the Author

Reviewer #1: Rethinking remdesivir for COVID-19:

a Bayesian reanalysis of trial findings

By: Sarahanne M. Field, Joyce M. Hoek, Ymkje Anna de Viries, Merie-Marie

Pittelkow, Maximilian Linde, Jasmine H. Muradchanian, Don van Ravenzwaaij

Submitted to: PLOS ONE

Ms I.d. PONE-D-20-30134

Report: 11/15/2020

Major Comments:

Recently remdesivir received much attention internationally for its treatment

effect against the COVID-19. The existing three studies on the effect of

remdesivir gave ambiguous conclusions. The authors use a Bayesian method to

reanalyze the results of the tree studies, using non-informative priors, and

summary statistics reported from the three previous studies.

Their reanalysis of the ACTT-1 trial data shows that remdesivir outperforms

placebo for time to clinical recovery. However, the evidence that remdesivir

for mortality rate is ambiguous.

Their reanalysis of Wang et al. provides weak evidence against

remdesivir-treated patients improving more quickly than patients in the

placebo group. The reanalysis of the mortality rate data yielded moderate

evidence in favor of no effect.

Their reanalysis of the GS-US-540-5773 trial data largely supports Goldman

and colleagues’ null hypothesis of no difference between a 5-or 10-day

course of remdesivir.

The authors finding is interesting. My comments are below.

* The authors should give explanation why a Bayesian method is preferred

than a frequentist one for this problem?

* The authors should try to provide the data links, so the data become

public accessible. Or at least provide the summary statistics of the

data as they used in this paper, either in the text or in an Appendix.

This will help the public researchers to evaluate performance of remdivir

and the that of the existing studies on remdivir.

* The authors mentioned that Jeffreys-Zellner-Siow Bayes factors based

on Bayesian t-test and chi-squared test were used in the reanalysis.

Please discuss the advantage of this method vs the classical Bayes

factor.

Minor Comments:

* The authors mentioned that non-informative priors are used in their

analysis. As there are several different non-informative priors, please

specify which non-informative prior is used.

Reviewer #2: This manuscript used Bayes factor to re-analyze the three trials on Remdesivir for treating COVID-19. The manuscript is well-written but has very limited contribution to either statistical method or clinical guidance. There is no discussion of why BF is used and how it improves the conclusions. There is no discussion of using and how to use different prior information. The authors can consider combining information from the three trials to improve conclusion.

6. PLOS authors have the option to publish the peer review history of their article (what does this mean?). If published, this will include your full peer review and any attached files.

Reviewer #1: **Yes: **Ao Yuan

Reviewer #2: No

- - - - -

---

## [Author Response · Author response to Decision Letter 0]

1 Apr 2021

Reviewer #1

R1-1 The authors should give explanation why a Bayesian method is preferred than a frequentist one for this problem?

In our opinion, there are many advantages to Bayesian hypothesis testing over frequentist hypothesis testing, such as the ability to continually test as data comes in (sequential testing) and quantifying the likelihood of hypotheses relative to one another. In the context of this specific reanalysis, the main advantage is the ability to quantify evidence in favor of the null hypothesis (i.e., the absence of efficacy), which traditional p-values do not allow for. 

When medicine regulators initially approved remdesivir, limited clinical trial data was available. In addition, one of the two available randomized placebo-controlled trials by Wang et al. could not be interpreted in a frequentist framework. Since we think it is important to include both trials in the regulatory efficacy assessment, we present a Bayesian reanalysis of the data. 

We agree that the original manuscript did not state explicitly why a Bayesian method is preferred in this specific case. We expanded our explanation of the added value of a Bayesian approach at lines 12-13, 19-25 of page 2 and lines 5-13 of page 3 of our manuscript. 

R1-2 The authors should try to provide the data links, so the data become public accessible. Or at least provide the summary statistics of the data as they used in this paper, either in the text or in an Appendix. 

All data and analysis code is available on the public OSF repository at this url: https://osf.io/kdqt3. We now also included this link in our manuscript at line 24 of page 3. 

R1-2: This will help the public researchers to evaluate performance of remdivir and the that of the existing studies on remdivir.

With regard to this second part of your comment, we wanted to facilitate the interpretation of our results by replacing the table of our results with a figure. This figure provides an overview of the current state of evidence for remdesivir in the treatment of COVID-19. The figure is presented at page 7 of our manuscript. We moved the table to the supplementary material. 

R1-3 The authors mentioned that Jeffreys-Zellner-Siow Bayes factors based on Bayesian t-test and chi-squared test were used in the reanalysis. Please discuss the advantage of this method vs the classical Bayes factor.

The Jeffreys-Zellner-Siow Bayes factor has as distinguishing property the use of the Cauchy prior, centered on zero. Citing from van Ravenzwaaij & Etz (in press, available at https://psyarxiv.com/27ndb/): 

"The Cauchy prior has some desirable mathematical properties (see e.g., Bayarri, Berger, Forte, & García-Donato, 2012; Consonni, Fouskakis, Liseo, & Ntzoufras, 2018), such as model selection consistency (for data generated under a model, the corresponding Bayes factor should go to infinity as sample size goes to infinity), predictive matching (a minimum sample size should exist for which the Bayes factor is 1, such that models are indistinguishable), and information consistency (a minimum sample size should exist for which data that result in test statistics that go to infinity should have corresponding Bayes factors that also go to infinity). Other priors may share some of these desirable properties, but the Cauchy prior has caught on as the go-to choice because it satisfies them all and is relatively easy to specify and interpret." 

We included an explanation on lines 3-8 of page 5 of our manuscript. 

R1-4 The authors mentioned that non-informative priors are used in their analysis. As there are several different non-informative priors, please specify which non-informative prior is used.

We now included a more extensive discussion of the methods that we used and our choice of priors under the subheadings ‘Bayesian two-sided t-test’ and ‘Bayesian chi-square test’ on page 4 and 5 of our manuscript and in our supplementary material. 

 

Reviewer #2

R2-1 This manuscript used Bayes factor to re-analyze the three trials on Remdesivir for treating COVID-19. The manuscript is well-written but has very limited contribution to either statistical method or clinical guidance. There is no discussion of why BF is used and how it improves the conclusions.

As indicated in our response to R1-1, our main reason for conducting the Bayesian reanalyses is to be able to quantify pro-null evidence. Under a frequentist framework, it remains unclear whether non-significant results (such as those by Wang et al.) are due to lack of power or due to null effects. Perhaps more importantly, concerns about "perceived significance" should not be a reason for rejection at PLOS ONE, and the reviewer does not present any compelling methodological concerns that could motivate a rejection.

In addition, we disagree that our paper has limited contribution to clinical guidance. When remdesivir was first approved, only two relevant clinical trials were available. One of these trials could not be interpreted under a frequentist framework and was excluded from regulatory assessments of remdesivir. When evidence is limited, it is important that all available evidence can be assessed. We provide a method to do so. 

As explained in our response to reviewer one, we did include a more explicit explanation of the added value of a Bayesian approach in this specific case in the introduction of our manuscript. 

R2-2 There is no discussion of using and how to use different prior information.

We have added detailed information about the choice of priors for both the Bayesian t-test and the Bayesian chi-square test to both the manuscript (page 4 and 5) and in the supplementary material. 

R2-3 The authors can consider combining information from the three trials to improve conclusion.

Thank you for your suggestion. We agree and are enthusiastic about combining information from the three trials, but as the data are quite different, we prefer to stick to combining the information qualitatively as we do in our current version of the manuscript. A (Bayesian) meta-analysis would not be justified, given the underlying assumptions. We explain our reasoning on lines 1-15 of page 4 of our manuscript.

---

## [Decision Letter · Decision Letter 1]

12 Jul 2021

Rethinking remdesivir for COVID-19: a Bayesian reanalysis of trial findings

PONE-D-20-30134R1

Dear Dr. Hoek,

We’re pleased to inform you that your manuscript has been judged scientifically suitable for publication and will be formally accepted for publication once it meets all outstanding technical requirements.

Kind regards,

Alan D Hutson

Academic Editor

PLOS ONE

Additional Editor Comments (optional):

Reviewers' comments:

Reviewer's Responses to Questions

**Comments to the Author**

1. If the authors have adequately addressed your comments raised in a previous round of review and you feel that this manuscript is now acceptable for publication, you may indicate that here to bypass the “Comments to the Author” section, enter your conflict of interest statement in the “Confidential to Editor” section, and submit your "Accept" recommendation.

Reviewer #1: All comments have been addressed

2. Is the manuscript technically sound, and do the data support the conclusions?

Reviewer #1: Partly

3. Has the statistical analysis been performed appropriately and rigorously? 

Reviewer #1: Yes

4. Have the authors made all data underlying the findings in their manuscript fully available?

Reviewer #1: Yes

5. Is the manuscript presented in an intelligible fashion and written in standard English?

Reviewer #1: Yes

6. Review Comments to the Author

Reviewer #1: Rethinking remdesivir for COVID-19:

a Bayesian reanalysis of trial findings

By: Sarahanne M. Field, Joyce M. Hoek, Ymkje Anna de Viries, Merie-Marie

Pittelkow, Maximilian Linde, Jasmine H. Muradchanian, Don van Ravenzwaaij

Submitted to: PLOS ONE

Ms I.d. PONE-D-20-30134-R1

Report: 05/21/2021

The authors addressed my comments. There are some parts still unclear.

* "the main advantage is the ability to quantify evidence in favor of the

null hypothesis". Do you mean that one can subjectively specify the prior

in favor of the null hypothesis? Is this an advantage or a way of imposing

personal opinion (which may be miss-leading)?

* "Since we think it is important to include both trials in the regulatory

efficacy assessment, we present a Bayesian reanalysis of the data".

As frequentist methods can also include multiple trials, please explain

what's unique for Bayesian on this point?

* The author mentioned that "the main advantage is the ability to quantify

evidence in favor of the null hypothesis", but they used a noninformative

prior, which does not favor the null nor the alternative. Please explain

how a noninformative prior can be in favor of the null hypothesis?

7. PLOS authors have the option to publish the peer review history of their article (what does this mean?). If published, this will include your full peer review and any attached files.

Reviewer #1: No

---

## [Editor Report · Acceptance letter]

16 Jul 2021

PONE-D-20-30134R1 

Rethinking Remdesivir for COVID-19: A Bayesian Reanalysis of Trial Findings 

Dear Dr. Hoek:

I'm pleased to inform you that your manuscript has been deemed suitable for publication in PLOS ONE. Congratulations! Your manuscript is now with our production department. 

Kind regards, 

on behalf of

Dr. Alan D Hutson 

Academic Editor

PLOS ONE